Repression of ZCT1, ZCT2 and ZCT3 affects expression of terpenoid indole alkaloid biosynthetic and regulatory genes

Li Chun Yao
Gibson Susan I. gibso043@umn.edu
Plant and Microbial Biology, University of Minnesota—Twin Cities , Saint Paul, MN , United States
Ma Wei
Electronic publication date: 2021 Jul 2
Publication date: 2021
Volume: 9
Electronic Location ID: e11624
Received 2021 Jan 25; Accepted 2021 May 26
Copyright: © 2021 Li and Gibson
Copyright year: 2021
Copyright holder: Li and Gibson
License: This is an open access article distributed under the terms of the Creative Commons Attribution License, which permits unrestricted use, distribution, reproduction and adaptation in any medium and for any purpose provided that it is properly attributed. For attribution, the original author(s), title, publication source (PeerJ) and either DOI or URL of the article must be cited.
License URL: https://creativecommons.org/licenses/by/4.0/

Keywords: Catharanthus roseus, ZCT1, ZCT2, ZCT3, GBF1, Terpenoid indole alkaloid, RNAi

Funding: United States National Science Foundation CBET-1064903 This work was supported by the United States National Science Foundation (CBET-1064903). The funders had no role in study design, data collection and analysis, decision to publish, or preparation of the manuscript.

==============================
Terpenoid indole alkaloids (TIAs) include several valuable pharmaceuticals. As Catharanthus roseus remains the primary source of these TIA pharmaceuticals, several research groups have devoted substantial efforts to increase production of these compounds by C. roseus. Efforts to increase TIA production by overexpressing positive regulators of TIA biosynthetic genes have met with limited success. This limited success might be due to the fact that overexpression of several positive TIA regulators turns on expression of negative regulators of TIA biosynthetic genes. Consequently, a more effective approach for increasing expression of TIA biosynthetic genes might be to decrease expression of negative regulators of TIA biosynthetic genes. Towards this end, an RNAi construct was generated that expresses a hairpin RNA carrying nucleotide fragments from three negative transcriptional regulators of TIA genes, ZCT1, ZCT2 and ZCT3, under the control of a beta-estradiol inducible promoter. Transgenic C. roseus hairy root lines carrying this ZCT RNAi construct exhibit significant reductions in transcript levels of all three ZCT genes. Surprisingly, out of eight TIA biosynthetic genes analyzed, seven (CPR, LAMT, TDC, STR, 16OMT, D4H and DAT) exhibited decreased rather than increased transcript levels in response to reductions in ZCT transcript levels. The lone exception was T19H, which exhibited the expected negative correlation in transcript levels with transcript levels of all three ZCT genes. A possible explanation for the T19H expression pattern being the opposite of the expression patterns of the other TIA biosynthetic genes tested is that T19H shunts metabolites away from vindoline production whereas the products of the other genes tested shunt metabolites towards vindoline metabolism. Consequently, both increased expression of T19H and decreased expression of one or more of the other seven genes tested would be expected to have similar effects on flux through the TIA pathway. As T19H expression is lower in the ZCT RNAi hairy root lines than in the control hairy root line, the ZCTs could act directly to inhibit expression of T19H. In contrast, ZCT regulation of the other seven TIA biosynthetic genes tested is likely to occur indirectly, possibly by the ZCTs turning off expression of a negative transcriptional regulator of some TIA genes. In fact, transcript levels of a negative TIA transcriptional regulator, GBF1, exhibited a strong, and statistically significant, negative correlation with transcript levels of ZCT1, ZCT2 and ZCT3. Together, these findings suggest that the ZCTs repress expression of some TIA biosynthetic genes, but increase expression of other TIA biosynthetic genes, possibly by turning down expression of GBF1.

Introduction

Catharanthus roseus (L) G. Don produces several terpenoid indole alkaloids (TIAs), including vincristine, vinblastine, ajmalicine and serpentine, that are widely prescribed chemotherapeutics. As production of these chemicals in microbes and chemical synthesis are currently not feasible (Van der Heijden et al., 2004; Shanks, 2005; Eastman, 2011), these chemicals continue to be harvested from the plant. Unfortunately, C. roseus produces these compounds in only very limited amounts. As a result, a number of research groups are characterizing the biosynthetic pathways leading to production of these chemicals and the regulatory pathways that help control flux through those pathways with the long-term goal of engineering plants to produce higher levels of these compounds.

The biochemical pathways leading to production of TIAs are complex (Fig. 1). The iridoid pathway catalyzes the synthesis of loganic acid from geranyl phosphate via multiple enzymatic steps (Miettinen et al., 2014). As part of this pathway, cytochrome P450 reductase (CPR) catalyzes the conversion of geraniol to 10-hydroxygeraniol (Meijer et al., 1993). The loganic acid is then converted to loganin via a reaction catalyzed by loganic acid-methyltransferase (LAMT) (Murata et al., 2008). A subsequent enzymatic reaction converts loganin to secologanin, one of the two precursors of TIA biosynthesis. The other precursor of TIA biosynthesis, tryptamine, is synthesized from tryptophan in a reaction catalyzed by tryptophan decarboxylase (TDC) (Goddijn et al., 1994). The enzyme strictosidine synthase (STR) then catalyzes the joining of secologanin and tryptamine to form strictosidine (Pasquali et al., 1992). Other TIAs are then synthesized from strictosidine via multiple, branching pathways. The enzymes 16-hydroxytabersonine-16-O-methyltransferase (16OMT) (Levac et al., 2008), desacetoxyvindoline 4-hydroxylase (D4H) (Vazquez-Flota et al., 1997) and deacetylvindoline acetyltransferase (DAT) (St-Pierre et al., 1998) catalyze reactions leading to the production of vindoline, which is then converted to vinblastine and vincristine via additional reactions. In contrast, tabersonine 19-hydroxylase (T19H) shunts metabolic flow away from production of vinblastine, by catalyzing the conversion of tabersonine to 19-hydroxytabersonine (Giddings et al., 2011).

Figure 1 TIA biosynthetic pathway in C. roseus.

The metabolites resulting from different biochemical reactions are indicated below the arrows. Solid arrows signify single enzymatic reactions whereas dotted arrows indicate multiple enzymatic reactions. The genes included for analysis are indicated next to the appropriate arrows.

Multiple transcriptional activators and repressors regulate the activity levels of TIA biosynthetic genes and genes from the TIA precursor pathways. ORCA2, an AP-2 domain protein, was the first transcriptional regulator of TIA biosynthetic genes identified from C. roseus. ORCA2 was originally shown to activate expression of STR (Menke et al., 1999). Overexpression of ORCA2 was subsequently shown to alter expression of several genes from the TIA biosynthetic pathway and both TIA precursor pathways (Li et al., 2013). A similar transcriptional regulator, ORCA3, was identified (Van der Fits & Memelink, 2000) and shown by several studies to alter expression of many of the genes in the TIA biosynthetic and TIA precursor pathways (Peebles et al., 2009; Pan et al., 2012; Schweizer et al., 2018). The CrBPF1 transcriptional activator was identified via a yeast one-hybrid screen using part of the STR promoter as bait (Van der Fits et al., 2000). However, overexpression of CrBPF1 has only modest effects on STR expression in C. roseus cell suspension cultures (Zhang et al., 2011) and has no significant effects in C. roseus hairy root cultures (Li et al., 2015). In contrast, overexpression of CrBPF1 in hairy root cultures does increase transcript levels for several genes from the TIA and both TIA feeder pathways (Li et al., 2015). The BIS1 transcriptional activator turns on expression of all genes analyzed that encode proteins in the metabolic pathway between geranyl diphosphate and loganic acid (Van Moerkercke et al., 2015; Van Moerkercke et al., 2016; Schweizer et al., 2018). The CrMYC1 transcriptional activator was identified based on its ability to bind the STR promoter (Chatel et al., 2003) and overexpression of CrMYC1 increases vinblastine, vincristine and catharanthine levels (Sazegari et al., 2018). However, information regarding which TIA and TIA-related genes may be regulated by CrMYC1 is currently lacking. In contrast, CrMYC2 has been shown to regulate the expression of several TIA biosynthetic and regulatory genes (Zhang et al., 2011; Schweizer et al., 2018). Similarly, overexpression of CrWRKY1 (Suttipanta et al., 2011) and CrWRKY2 (Suttipanta, 2011) in C. roseus hairy root cultures has been shown to affect transcript levels of several TIA biosynthetic and regulatory genes.

In addition to TIA transcriptional activators, several genes encoding TIA transcriptional repressors have been identified. GBF1 and GBF2 are transcriptional repressors that bind to elements in the STR and TDC gene promoters (Pré et al., 2000; Sibéril et al., 2001). ZCT1, ZCT2 and ZCT3 are TIA transcriptional repressors that were identified via a yeast one-hybrid screen using part of the TDC promoter as bait (Pauw et al., 2004). A comparison of the amino acid sequences of the ZCTs revealed that they share several motifs, including an L-box, B-box and LxLxL sequences (Pauw et al., 2004). All three ZCT proteins bind the TDC and STR promoters, although the exact binding sites within these promoters are different for ZCT1 and ZCT2 than for ZCT3. ZCT1, ZCT2 and ZCT3 suppress STR and TDC promoter activities in transactivation experiments and can also suppress the activities of ORCA2 and ORCA3 on the STR promoter (Pauw et al., 2004; Mortensen et al., 2019a). However, decreasing expression of ZCT1 and ZCT2 using an RNAi construct designed to reduce ZCT1 expression did not reduce expression of TDC, STR or G10H (Rizvi et al., 2016). ZCT1 and ZCT2, but not ZCT3, also repress expression of HDS, part of the methyl erythritol pathway (Chebbi et al., 2014). ZCT1 has also been shown to repress its own promoter (Mortensen et al., 2019b).

Several research groups have attempted to increase production of valuable TIAs by overexpressing TIA transcriptional activators. However, these efforts have generally met with limited success. For example, overexpression of ORCA2 led to initial increases in transcript levels of PRX1, a major class III peroxidase that catalyzes the formation of α-3′,4′-anhydrovinblastine (Costa et al., 2008). However, these increases were transient. In addition, overexpression of ORCA2 caused dramatic decreases, rather than increases, in DAT transcript levels (Li et al., 2013). A possible explanation for these findings is that some TIA transcriptional activators induce expression of TIA transcriptional repressors which then turn down expression of specific TIA biosynthetic genes. Consistent with this possibility are findings that overexpression of several TIA transcriptional activators turns on expression of TIA transcriptional repressors. For example, overexpression of either ORCA3 (Peebles et al., 2009) or ORCA2 (Li et al., 2013) causes increases in the transcript levels of all three ZCT genes, but has no significant effects on expression of GBF1 or GBF2. Similarly, CrBPF1 overexpression increases transcript levels of 11 out of 13 TIA transcriptional regulators assayed, including all three ZCT genes and both GBF genes, although the effects on GBF2 expression are relatively minor (Li et al., 2015). Overexpression of CrWRKY1 causes increased expression of all three ZCT genes, but not of GBF1 nor GBF2 (Suttipanta et al., 2011). In contrast, overexpression of CrWRKY2 in transgenic hairy roots causes substantial increases in expression of ZCT1 and ZCT3, but has only minor effects on expression of ZCT2, GBF1 and GBF2 (Suttipanta, 2011).

Findings that overexpression of TIA transcriptional activators often results in increased expression of TIA transcriptional repressors led to the suggestion that increased production of TIAs may require mechanisms designed to decrease expression of TIA repressors (Peebles et al., 2009). In an attempt to do this, researchers transformed C. roseus with an RNAi construct designed to turn off expression of ZCT1. This RNAi construct turns down expression of ZCT2 in addition to ZCT1, but has no significant effects on ZCT3 expression. C. roseus lines with this RNAi construct have no significant alterations in TIA levels, suggesting that reduced expression of all three ZCT genes may be necessary to increase TIA levels (Rizvi et al., 2016). Towards that end, in this study an RNAi construct designed to decrease expression of all three ZCT genes was designed and used to generate transgenic C. roseus hairy root lines.

Materials & methods

Plant materials and growth conditions

Catharanthus roseus, Vinca Little Bright Eye (www.neseed.com), was used for all of the experiments described. Seeds were placed in 50 mL falcon tubes and a solution consisting of 50% bleach and 0.02% triton X-100 was added to the seeds. The seeds were left in the solution for 7 to 8 min. The seeds were then washed with sterile water. Next, the seeds were germinated on Gamborg’s B5 medium (Sigma, St. Louis, MO, USA) supplemented with Gamborg’s vitamins (Sigma, St. Louis, MO, USA). Seeds were allowed to germinate in the dark for 2 weeks at 26 °C. The seedlings were then shifted to a 16-h-light/8-h-dark cycle with a light intensity of approximately 44 µmol m−2 s−1. After approximately 4 weeks of additional growth time, the seedlings were inoculated with Agrobacterium tumefaciens. The genes analyzed in this work are: 16OMT (GenBank: EF444544), BIS1 (GenBank: KM409646), CPR (GenBank: X69791), CrBPF1 (GenBank: AJ251686), CrMYC1 (GenBank: AF283506), CrMYC2 (GenBank: AF283507), CrWRKY1 (GenBank: HQ646368), CrWRKY2 (GenBank: JX241693), D4H (GenBank: U71605), DAT (GenBank: AF053307), EF-1 (GenBank: EU007436), GBF1 (GenBank: AF084971), GBF2 (GenBank: AF084972), LAMT (GenBank: EU057974), ORCA2 (GenBank: AJ238740), ORCA3 (GenBank: EU072424), STR (GenBank: X53602), T19H (GenBank: HQ901597), TDC (GenBank: X67662), UBQ11 (GenBank: EU007433), ZCT1 (GenBank: AJ632082), ZCT2 (GenBank: AJ632083) and ZCT3 (GenBank: AJ632084).

Generation of a beta-estradiol inducible RNAi construct

The pOpOff2(Hyg) RNAi vector, generously provided by CSIRO (Wielopolska et al., 2005), was modified by replacement of the dexamethasone-inducible promoter with a beta-estradiol inducible promoter from the pER8 XVE inducible system (Zuo, Niu & Chua, 2000). The resulting vector was designed as the XVE-pOpOff2 RNAi vector. Fragments of the ZCT1, ZCT2 and ZCT3 genes that were 320–338 bp in length were amplified using KOD Hot Start DNA polymerase (Novagen, Madison, WI, USA) and genomic DNA from C. roseus variety Little Bright Eye. The oligonucleotides used for these PCR reactions were as follows: ZCT1, 5′ TGGAGAAATCAGCAGTACCGGCGT 3′ paired with 5′ TGGTACCGCCTTTGCAACAGG 3′; ZCT2, 5′ TACCGATGAAGCGTACGAGA 3′ paired with 5′ ACCTCCGAGAGCTTGACCGATAGC 3′ and ZCT3, 5′ ACGAAAACGCAGCTACTCTCCGCT 3′ paired with 5′ TGCCTTATGTCCTCCGAGTGCTTGG 3′. Oligonucleotides 5′ GTTGCAAAGGCGGTACCATACCGATGAAGCGTACGAG 3′ and 5′ GGTCAAGCTCTCGGAGGTACGAAAACGCAGCTACTC 3′ were used to run a bridge PCR to combine the ZCT1, ZCT2 and ZCT3 DNA fragments into one larger nucleotide fragment. The ZCT1/2/3 fragment was cloned into the PCR8/GW/topo entry vector (Invitrogen, Grand Island, NY, USA) and then transferred to the XVE-pOpOff2 RNAi vector through the LR reaction using LR clonase mix II (Invitrogen, Grand Island, NY, USA). The resulting construct, designated XVE-pOpOff2-ZCT (XPZ), contains two copies of the ZCT1/2/3 DNA fragment. The two ZCT1/2/3 sequences are in an inverted repeat orientation and are separated by the DNA spacer present in the original pOpOff2(Hyg) RNAi vector (Fig. S1). The XPZ construct was then used for transformation of Agrobacterium tumefaciens strain GV3101.

Generation of transgenic hairy roots

Approximately 6-weeks old C. roseus seedlings were used for the plant transformation procedure. Transformation experiments were carried out as previously described (Li et al., 2013). In particular, transformation was achieved using an approximately equal mixture of A. tumefaciens strain GV3101 cultures transformed with the XPZ RNAi construct or with the pPZPROL plasmid. The pPZPROL plasmid carries the rol ABC genes, which have been shown to be sufficient to induce formation of hairy roots on C. roseus. A. tumefaciens strain GV3101 carrying the rol ABC genes was used for these experiments because the hairy roots produced in this way tend to show better adaptability to growth in liquid culture than those produced using A. rhizogenes (Hong et al., 2006). Hairy roots appeared on inoculation sites approximately 4 weeks later. After the hairy roots reached lengths of approximately 1 cm, they were excised and transferred to solid medium supplemented with 30 g L−1 sucrose, 6 g L−1 agar, 250 mg L−1 cefotaxime, half-strength Gamborg’s B5 salts and full-strength Gamborg’s vitamins (pH 5.8). After 1 week of growth on solid media, 30 mg L−1 hygromycin was used to select for hairy roots carrying the XPZ construct. Hairy roots carrying the XPZ construct were transferred to 50 mL of liquid media, comprised of 50 mL of half-strength Gamborg’s B5 liquid solution supplemented with full-strength Gamborg’s vitamins and 30 g L−1 sucrose. The flasks containing the transgenic hairy roots were kept on a shaker at 225 rpm in the dark and were sub-cultured every 5 weeks. To confirm that these hairy roots carried the XPZ RNAi construct, pcr was used to amplify sequences from the ZCT1 to ZCT3 fragments carried on the XPZ RNAi construct, from the T-DNA right border to the ZCT1 fragment and from the ZCT3 fragment to the XVE sequences. The results of these pcr experiments confirmed the presence of the XPZ RNAi construct (data not shown). To generate a negative control line, C. roseus was also transformed with XVE-pOpOff2 empty vector, using the same procedure described above for transformation with the XPZ RNAi construct.

Induction of transgene expression and tissue collection

Induction of expression of the RNAi construct was carried out was carried out largely as described previously (Li et al., 2013). Specifically, to induce expression of the ZCT1/2/3 hairpin sequence, three actively growing hairy roots, each 3 to 4 cm in length, were transferred to a 250 mL flask containing 50 mL of half-strength B5 media. The cultures were grown on a shaker at 100 rpm in the dark for 31 days. The media was replaced with fresh half-strength B5 on days 17 and 28. On day 31 expression of the RNAi construct was induced by addition of 50 µL of 20 mM beta-estradiol (in ethanol) to the liquid culture, for a final concentration of 20 µM beta-estradiol. As a control, un-induced cultures were treated with 50 µL of ethanol at the same time. The hairy root cultures were returned to a dark environment. Cultures were harvested 0, 6, 12, 24, 48 and 72 h after the start of induction. Hairy roots transformed with the XVE-pOpOff2 empty vector were used as a negative control and were treated the same way as the hairy roots transformed with the XPZ RNAi construct. Three independent hairy root cultures were harvested for each transgenic hairy root line, time point and media combination. Upon collection, hairy root samples were immediately flash frozen in liquid nitrogen and then stored at −80 °C prior to being used for gene expression analyses.

RNA extraction and RT-qPCR analyses

Total RNA was extracted as previously described (Li et al., 2013) using the Spectrum Total RNA Isolation Kit (Sigma, St. Louis, MO, USA) with on-column DNase I digestion. cDNAs were synthesized using goscript reverse transcriptase (Promega, Madison, WI, USA). Gene transcript levels were analyzed by qPCR using the SYBR Premix EX Taq II (2X) (Tli RNase H plus (Clontech Laboratories, Mountain View, CA, USA)) and reaction were run on a Roche LightCycler 480 II. For all experiments qPCR data were normalized using the geometric average of qPCR results for two control genes, EF1 and UBQ11, which were shown previously to be the two most stably expressed genes of those tested in C. roseus (Wei, 2010). For statements of fold changes in transcript levels, a change of one Ct (i.e., a change of one PCR cycle) was estimated to represent a two-fold change in transcript levels. Relative mRNA levels are expressed as ∆∆Ct. ∆∆Ct = ∆Ctun-induced control line at 0 h − ∆Ctother. ∆Ctun-induced control line at 0 h = Ctindicated gene in un-induced control line at 0 h – CTEF1/UBQ11in un-induced control line at 0 h. ∆Ctother = Ctindicated gene – CtEF1/UBQ11 for the time point, line and growth condition being analyzed.

Statistical analyses

A two-tailed Student’s T-test was employed to determine statistical significance between induced and un-induced cultures of the same hairy root line at the same timepoint, or between an RNAi line and the control line grown on the same media for the same amount of time. The Pearson product moment correlation coefficient was used to identify correlations between expression levels of different pairs of genes across all hairy root lines, culture conditions and timepoints analyzed. The significance of the Pearson product moment correlation coefficient was determined using the p value calculator at: https://www.socscistatistics.com/pvalues/pearsondistribution.aspx.

Results and discussion

ZCT expression is significantly decreased in RNAi hairy root lines

The ZCT1, ZCT2 and ZCT3 transcriptional regulators have been shown to act as negative regulators of genes involved in TIA metabolism and regulation of TIA metabolism (Pauw et al., 2004; Chebbi et al., 2014; Mortensen et al., 2019a; Mortensen et al., 2019b). To investigate the role of the ZCTs in these processes, it was desirable to obtain C. roseus hairy root lines with decreased expression of all of the ZCTs. Towards this end, an RNAi construct was generated that is designed to reduce expression of all three of the ZCT genes. This RNAi construct, designated XPZ, carries inverted repeats of a DNA sequence that contains 320–338 bp fragments from each of the three ZCT genes. These sequences are designed to be expressed under the control of a beta-estradiol inducible promoter (Zuo, Niu & Chua, 2000) in the XVE-pOpOff2 vector. An inducible promoter was used for these experiments as there was a concern that constitutive repression of the ZCT genes could be harmful to the growth of transgenic hairy roots. The XPZ construct was used to generate C. roseus transgenic hairy root lines. Based on preliminary testing of transgene expression levels, two lines (XPZ28 and XPZ38) were chosen for further analysis. A negative control line (C) was generated using the XVE-pOpOff2 empty vector.

To test the effects of the XPZ construct on ZCT gene expression, two hairy root lines carrying the XPZ construct (XPZ28 and XPZ38) and one control line (C) were grown in the presence or absence of the inducer of the XPZ RNAi construct, beta-estradiol. Cultures were harvested 0, 6, 12, 24, 48 and 72 h after addition of 0 or 20 µM beta-estradiol to the cultures. Analysis of ZCT1, ZCT2 and ZCT3 mRNA levels in tissues harvested from these cultures revealed that expression of all three genes was significantly reduced in the XPZ28 and XPZ38 lines compared to the control line (Fig. 2). At the zero timepoint, ZCT1 expression was down 13 fold in the XPZ28 line relative to the control line and 29 fold in the XPZ38 line (Fig. 2A). Similarly, expression of ZCT2 was down 39 fold and 34 fold in the XPZ28 and XPZ38 lines, respectively (Fig. 2B). Expression of the ZCT3 gene was decreased to a lesser, but still significant extent, being down six and four fold in the XPZ28 and XPZ38 lines, respectively (Fig. 2C). These results indicate that the RNAi construct causes a significant reduction in expression of all three ZCT genes, even in the absence of the inducing agent for the construct. The most likely explanation for this finding is that expression of the RNAi construct may be leaky, with substantial expression occurring even in the absence of the inducing agent. The transgenic RNAi lines grew well, alleviating concerns that constitutive repression of ZCT expression could have deleterious effects on growth of hairy root cultures.

Figure 2 C. roseus transgenic hairy root lines carrying the XPZ RNAi construct have decreased ZCT transcript levels.

(A) ZCT1, (B) ZCT2 and (C) ZCT3 transcript levels were assayed in the XPZ28 (28) and XPZ38 (38) transgenic hairy root lines, which carry the XPZ RNAi construct designed to reduce expression of all three ZCT genes. ZCT transcript levels were also analyzed in a control line (C), transformed with the XVE-pOpOff2 empty vector. Cultures from all three hairy root lines were treated with 0 µM (0) or 20 µM (20) beta-estradiol and then harvested after the indicated number of hours. The results of Student’s T-tests, comparing different lines or treatment conditions at different time points, are shown below each graph. T-test results below 0.05 are highlighted by bold font. Results are the average ∆∆CT value of three biological replicates, with two technical replicates per biological replicate. Error bars indicate standard deviations.

Although expression of the ZCT genes was significantly decreased in both RNAi hairy root lines prior to the addition of beta-estradiol, growth on beta-estradiol did result in further reductions in expression of all three ZCT genes. For example, at the 12-h timepoint ZCT1 expression was down approximately seven fold (Fig. 2A), ZCT2 expression was down approximately four fold (Fig. 2B) and ZCT3 expression was down three to four fold (Fig. 2C) in the RNAi lines grown on 20 µM beta-estradiol relative to the same lines grown on 0 µM beta-estradiol. When the RNAi lines growing on 20 µM beta-estradiol were compared with the control line growing on 20 µM beta-estradiol, the reductions in ZCT expression in the RNAi lines were quite large and were statistically significant at all timepoints tested. For example, after growth on 20 µM beta-estradiol for 12 h, expression of all three ZCT genes was down by approximately 20 to 80 fold in both RNAi lines relative to the control line.

Decreased expression of the ZCT genes results in altered expression of TIA biosynthetic genes

As the ZCT genes have been shown to act as negative regulators of transcription, decreased expression of the ZCT genes was expected to result in increased expression of at least some TIA biosynthetic and regulatory genes. To test this hypothesis, transcript levels of genes from both the monoterpenoid and indole feeder pathways and from the TIA pathways were analyzed. A time course was performed for these experiments as several previous studies have shown that the effects of altering expression of regulatory genes on expression of TIA biosynthetic and regulatory genes are often transitory (Costa et al., 2008; Peebles et al., 2009; Li et al., 2013, 2015). Therefore, it was desirable to assay several timepoints to identify as many regulated genes as possible. However, the leakiness of the beta-estradiol promoter resulted in substantial levels of ZCT repression even in the absence of the inducing agent. In fact, as seen below, many of the genes found to be regulated by the ZCTs exhibit approximately equal levels of regulation at the 0 h timepoint as at later timepoints (i.e., after addition of the inducing agent). These results suggest that many of the observed effects might be due to long-term repression of ZCT expression. It is possible that short-term repression of ZCT expression could have somewhat different effects on expression of TIA regulatory and biosynthetic genes.

Surprisingly, expression of the CPR (Fig. 3A) and LAMT (Fig. 3B) genes from the monoterpenoid pathway was found to be reduced, rather than increased, in the RNAi lines relative to the control line. Although the decreases in CPR transcript levels were quite modest, reaching a maximum of a five-fold reduction in the XPZ38 line relative to the control line after growth on 20 µM beta-estradiol for 48 h, they were statistically significant at most timepoints analyzed for the XZP38 line (Fig. 3A). In addition, when comparing CPR expression levels with ZCT1, ZCT2 or ZCT3 expression levels in all hairy root lines, timepoints and media analyzed, a statistically significant positive correlation was found between CPR expression and expression of each of the ZCT genes (Table 1). Similarly, LAMT transcript levels were reduced by approximately three to four fold in the RNAi lines compared to the control line at the same timepoint and beta-estradiol concentration (Fig. 3B). Determination of the Pearson product moment correlation coefficient for LAMT expression and expression of each of the ZCT genes also revealed a statistically significant positive correlation between LAMT expression and expression of each of the ZCT genes (Table 1).

Figure 3 Time course of CPR and LAMT expression.

(A) CPR and (B) LAMT transcript levels were assayed in the XPZ28 (28), XPZ38 (38) and control lines (C) grown on 0 µm (0) or 20 µM (20) beta-estradiol and harvested at the indicated time points. The results of Student’s T-tests, comparing different lines or treatment conditions at different time points, are shown below each graph. T-test results below 0.05 are highlighted by bold font. Results are the average ∆∆CT value of three biological replicates, with two technical replicates per biological replicate. Error bars indicate standard deviations.

Table 1 Correlations between expression of ZCT1, ZCT2 and ZCT3 and expression of TIA biosynthetic and regulatory genes.

Genes being compared	Pearson correlation coefficient	P value	
	ZCT1	ZCT2	ZCT3	ZCT1	ZCT2	ZCT3	
CPR	0.62	0.49	0.36	0.00014	0.0039	0.039	
LAMT	0.75	0.88	0.85	<0.00001	<0.00001	<0.00001	
TDC	0.73	0.66	0.52	<0.00001	0.000033	0.0022	
STR	0.57	0.56	0.54	0.00048	0.00063	0.0011	
T19H	−0.47	−0.48	−0.44	0.0064	0.0043	0.0098	
16OMT	0.61	0.59	0.42	0.00018	0.00032	0.015	
D4H	0.77	0.77	0.68	<0.00001	<0.00001	0.000014	
DAT	0.85	0.93	0.86	<0.00001	<0.00001	<0.00001	
ORCA2	−0.27	−0.29	−0.44	0.12	0.11	0.011	
ORCA3	−0.20	−0.47	−0.25	0.28	0.0060	0.17	
CrBPF1	0.24	0.35	0.32	0.18	0.046	0.067	
BIS1	0.72	0.77	0.59	<0.00001	<0.00001	0.00031	
CrMYC1	0.65	0.54	0.47	0.000050	0.0011	0.0060	
CrMYC2	−0.02	−0.15	0.02	0.92	0.40	0.93	
CrWRKY1	0.36	0.35	0.37	0.041	0.046	0.035	
CrWRKY2	0.36	0.30	0.14	0.039	0.091	0.45	
GBF1	−0.59	−0.78	−0.73	0.00033	<0.00001	<0.00001	
GBF2	0.62	0.56	0.55	0.00012	0.00080	0.00090	
ZCT1	N/A	0.91	0.92	N/A	<0.00001	<0.00001	
ZCT2	0.91	N/A	0.92	<0.00001	N/A	<0.00001	
ZCT3	0.92	0.92	N/A	<0.00001	<0.00001	N/A	
Note:

The Pearson product moment correlation coefficient was determined for the transcript levels of each gene analyzed with the transcript levels of the ZCT1, ZCT2 or ZCT3 genes in the same hairy root line grown for the same amount of time on the same media. P values indicate the significance of the Pearson product moment correlation coefficient (N = 33), with values below 0.05 considered to be statistically significant. Bold font indicates a statistically significant positive correlation in gene expression between the two genes being compared. Italicized font indicates a statistically significant negative correlation in gene expression between the two genes being compared.

TDC catalyzes the last step in the indole pathway leading to production of one of the two TIA precursors, tryptamine (Goddijn et al., 1994). On average, decreased expression of the ZCT genes in the RNAi lines caused a small reduction in TDC transcript levels (Fig. 4). Reductions in TDC transcript levels were particularly modest in the XPZ28 line, being statistically significant at only the 6- and 12-h timepoints in the cultures growing on 0 µM beta-estradiol and at the 12-h timepoint for the cultures growing on 20 µM beta-estradiol. The XPZ38 line exhibited a greater reduction in TDC transcript levels. TDC expression was significantly decreased in the XPZ38 line on both media and at all timepoints tested, with the exception of the 24-h timepoint for cultures growing on 20 µM beta-estradiol (Fig. 4). Although TDC expression was decreased less than two fold on average in the XPZ28 line and between two and three fold in the XPZ38 line, analysis of the Pearson correlation coefficient for TDC expression levels and ZCT1, ZCT2 or ZCT3 expression levels revealed that there is a statistically significant positive correlation between TDC expression and expression of all three ZCT genes (Table 1).

Figure 4 Time course of TDC expression.

TDC transcript levels were assayed in the XPZ28 (28), XPZ38 (38) and control lines (C) grown on 0 µm (0) or 20 µM (20) beta-estradiol and harvested at the indicated time points. The results of Student’s T-tests, comparing different lines or treatment conditions at different time points, are shown below each graph. T-test results below 0.05 are highlighted by bold font. Results are the average ∆∆CT value of three biological replicates, with two technical replicates per biological replicate. Error bars indicate standard deviations.

To determine whether reductions in ZCT expression levels can affect expression of TIA biosynthetic genes, transcript levels for several TIA biosynthetic genes were analyzed in the ZCT RNAi and control lines (Fig. 5). STR catalyzes the first step in TIA biosynthesis, namely the combination of secologanin and tryptamine to form strictosidine (Pasquali et al., 1992). Decreased expression of the ZCT genes had a minor, negative effect on STR transcript levels. Reductions in STR transcript levels were statistically significant at only a few of the combinations of hairy root lines, media and timepoints tested (Fig. 5A). However, analysis of the Pearson correlation coefficient for STR expression levels and ZCT1, ZCT2 or ZCT3 expression levels revealed that there is a statistically significant positive correlation between STR expression and expression of all three of the ZCT genes (Table 1). T19H catalyzes the conversion of tabersonine to 19-hydroxytabersonine (Giddings et al., 2011). Interestingly, decreased expression of the ZCT genes caused an average increase in T19H expression (Fig. 5B). Increases in T19H transcript levels averaged only approximately 50% for both RNAi lines, and were statistically significant for only a few of the combinations of hairy root line, media and timepoint tested. However, there was a statistically significant negative correlation between T19H transcript levels and expression levels of all three ZCT genes when analyzing all of the hairy root lines, media and timepoints analyzed (Table 1). In contrast, reductions in ZCT expression levels caused a decrease in 16OMT expression levels (Fig. 5C). 16OMT catalyzes the conversion of 16-hydroxytabersonine to 16-methoxytabersonine (Levac et al., 2008). 16OMT expression levels were down an average of 1.5 fold in the XPZ28 line and 2.5 fold in the XPZ38 line. Analysis of the Pearson correlation coefficient for 16OMT expression and expression of the ZCT1, ZCT2 or ZCT3 genes revealed a statistically significant positive correlation between 16OMT and ZCT expression levels. Similarly, expression of the D4H gene was down an average of approximately two or three fold in the XPZ28 and XPZ38 lines, respectively, relative to the control line (Fig. 5D). D4H catalyzes the conversion of desacetoxyvindoline to deacetylvindoline (Vazquez-Flota et al., 1997). Decreased expression of the ZCT genes also resulted in an average decrease of approximately four fold in DAT expression in both RNAi lines (Fig. 5E). DAT catalyzes the conversion of deacetylvindoline to vindoline (St-Pierre et al., 1998). Analyses of Pearson correlation coefficients between expression of D4H or DAT and ZCT1, ZCT2 or ZCT3 revealed a statistically significant positive correlation between expression of D4H and DAT and expression of each of the ZCT genes (Table 1).

Figure 5 Time course of expression of TIA biosynthetic genes.

(A) STR, (B) T19H, (C) 16OMT, (D) D4H and (E) DAT transcript levels were assayed in the XPZ28 (28), XPZ38 (38) and control lines (C) grown on 0 µm (0) or 20 µM (20) beta-estradiol and harvested at the indicated time points. The results of Student’s T-tests, comparing different lines or treatment conditions at different time points, are shown below each graph. T-test results below 0.05 are highlighted by bold font. Results are the average ∆∆CT value of three biological replicates, with two technical replicates per biological replicate. Error bars indicate standard deviations.

As the ZCT genes have previously been characterized as negative transcriptional regulators, decreased expression of the ZCT genes was anticipated to lead to increased expression of at least some TIA biosynthetic genes. Surprisingly, seven of the eight biosynthetic genes tested exhibited decreased rather than increased expression in response to reductions in ZCT transcript levels. The lone exception was T19H, which exhibited the expected negative correlation in transcript levels with ZCT transcript levels. A possible explanation for the T19H expression pattern being the opposite of the expression patterns of the other biosynthetic genes tested is that T19H shunts metabolites away from vindoline production whereas the other genes tested shunt metabolites towards vindoline metabolism (Fig. 1). Consequently, both increased expression of T19H and decreased expression of one or more of the other seven genes tested would be expected to have similar effects on flux through different branches of the TIA pathway. For example, both increased expression of T19H and decreased expression of one of the other seven TIA biosynthetic genes would be expected to result in decreased flux from tabersonine to vindoline.

Decreased expression of the ZCT genes results in altered expression of TIA regulatory genes

Surprisingly, decreased expression of the ZCT genes led to decreased expression of seven of the eight biosynthetic genes analyzed (Fig. 5). As the ZCT genes have previously been characterized as negative regulators of transcription, a possible explanation for this result is that the ZCTs affect expression of some of the TIA biosynthetic genes indirectly. For example, it is possible that the ZCTs cause decreased expression of one or more of the other negative regulators of expression of TIA biosynthetic genes. In that case, decreased expression of the ZCT genes could lead to increased expression of those negative regulator(s) and, consequently, decreased expression of the biosynthetic genes regulated by those other negative regulators. As decreased expression of the ZCT genes led to increased expression of T19H, the ZCTs could be affecting T19H expression directly or indirectly. Direct repression of T19H expression could be caused, for example, by one or more of the ZCTs binding to a regulatory sequence in the T19H promoter and interfering with transcription of T19H. Alternatively the ZCTs could affect T19H expression levels indirectly by repressing expression of a positive regulator of T19H expression.

Additional analyses were done to test these possible mechanisms for ZCT regulation of TIA biosynthetic genes. The expression levels of eight genes (BIS1, CrBPF1, CrMYC1, CrMYC2, CrWRKY1, CrWRKY2, ORCA2 and ORCA3) previously implicated as positive TIA transcriptional regulators and of two genes (GBF1 and GBF2) previously implicated as negative TIA transcriptional regulators were determined. Expression of ZCT3, but not of ZCT1 or ZCT2, exhibited a statistically significant negative correlation with expression of ORCA2. In contrast, ZCT2, but not ZCT1 or ZCT3, exhibited a statistically significant negative correlation with expression of ORCA3 (Table 1). However, the effects of ZCT3 and ZCT2 on expression of ORCA2 (Fig. 6A) and ORCA3 (Fig. 6B) were of limited magnitude. BIS1 transcript levels are lower in the RNAi lines than in the control line at most timepoints tested (Fig. 6C). Although this is especially true for the XPZ38 line, even the XPZ38 exhibited only modest differences in BIS1 transcript levels relative to the control line. However, expression of all three ZCT genes exhibited statistically significant positive correlations with BIS1 transcript levels (Table 1). In the case of CrBPF1, ZCT expression levels generally had little effect on CrBPF1 transcript levels (Fig. 6D), although ZCT2 transcript levels did exhibit a statistically significant positive correlation with CrBPF1 transcript levels (Table 1). Decreased ZCT expression tended to lead to decreased CrMYC1 expression, especially for the XPZ38 line grown on media containing 20 µM beta-estradiol (Fig. 6E). In addition, expression levels of all three ZCT genes showed statistically significant positive correlations with CrMYC1 transcript levels (Table 1). In contrast, ZCT expression levels had very little effect on CrMYC2 transcript levels (Fig. 6F) and there were no statistically significant correlations between CrMYC2 transcript levels and the transcript levels of any of the ZCT genes (Table 1). ZCT expression levels exhibited minor and inconsistent effects on CrWRKY1 transcript levels, although there was a tendency for CrWRKY1 transcript levels to be slightly decreased in the RNAi lines (Fig. 6G) and there was a statistically significant positive correlation between CrWRKY1 transcript levels and transcript levels of each of the ZCT genes (Table 1). CrWRKY2 transcript levels tended to be down in the XPZ38 line, but exhibited inconsistent results for the XPZ28 line (Fig. 6H). Transcript levels of ZCT1, but not of ZCT2 or ZCT3, exhibited a statistically significant positive correlation with CrWRKY2 transcript levels (Table 1).

Figure 6 Time course of expression of TIA positive transcriptional regulators.

(A) ORCA2, (B) ORCA3, (C) BIS1, (D) CrBPF1, (E) CrMYC1, (F) CrMYC2, (G) CrWRKY1 and (H) CrWRKY2 transcript levels were assayed in the XPZ28 (28), XPZ38 (38) and control lines (C) grown on 0 µm (0) or 20 µM (20) beta-estradiol and harvested at the indicated time points. The results of Student’s T-tests, comparing different lines or treatment conditions at different time points, are shown below each graph. T-test results below 0.05 are highlighted by bold font. Results are the average ∆∆CT value of three biological replicates, with two technical replicates per biological replicate. Error bars indicate standard deviations.

The strongest effects on expression of a TIA transcriptional regulator were observed for the negative transcriptional regulator, GBF1. In line XPZ28, GBF1 transcript levels were typically two- to 10-fold higher than in the control line at the same timepoint and grown on the same media. For line XPZ38, GBF1 transcript levels were typically 10- to 30-fold higher than in the control line at the same timepoint and grown on the same media (Fig. 7A). In addition, the transcript levels of all three ZCT genes exhibited strong, statistically significant negative correlations with GBF1 transcript levels (Table 1). Interestingly, ZCT transcript levels had the opposite effect on expression of GBF2 as on expression of GBF1, as GBF2 transcript levels tended to be slightly lower in the RNAi lines than in the control line grown on the same media (Fig. 7B). Although the effects of ZCT transcript levels on GBF2 transcript levels were relatively minor, the expression levels of all three ZCT genes did exhibit statistically significant positive correlations with GBF2 transcript levels (Table 1). However, because GBF2 transcript levels were down an average of only approximately 12% in the induced XPZ28 cultures and 26% in the induced XPZ38, cultures, GBF1 is likely to play a much more important role in regulation of TIA biosynthetic gene expression than does GBF2.

Figure 7 Time course of expression of TIA negative transcriptional regulators.

(A) GBF1 and (B) GBF2 transcript levels were assayed in the XPZ28 (28), XPZ38 (38) and control lines (C) grown on 0 µm (0) or 20 µM (20) beta-estradiol and harvested at the indicated time points. The results of Student’s T-tests, comparing different lines or treatment conditions at different time points, are shown below each graph. T-test results below 0.05 are highlighted by bold font. Results are the average ∆∆CT value of three biological replicates, with two technical replicates per biological replicate. Error bars indicate standard deviations.

The positive correlations observed between ZCT transcript levels and transcript levels of seven of the eight TIA biosynthetic genes analyzed could be explained by the ZCTs negatively regulating expression of one or more of the other negative regulators of TIA biosynthetic genes. Besides the ZCTs, GBF1 and GBF2 have been postulated to act as negative regulators of some TIA biosynthetic genes (Pré et al., 2000; Sibéril et al., 2001). Interestingly, GBF1 transcript levels were found to be much higher in both of the ZCT RNAi lines than in the control line and the Pearson product moment correlation coefficient showed a statistically significant negative correlation for transcript levels for all three of the ZCT genes and GBF1 transcript levels (Table 1). These findings suggest a model where all three of the ZCTs act as negative regulators of GBF1 expression and GBF1, in turn, acts as a negative regulator of expression of CPR, LAMT, TDC, STR, 16OMT, D4H and DAT. The available data do not indicate whether the effects of the ZCTs on GBF1 expression, and the effects of GBF1 on expression of the TIA biosynthetic genes, are direct or indirect. For example, the ZCTs could bind to elements in the GBF1 promoter and directly repress expression of GBF1. Alternatively, the ZCTs could reduce GBF1 expression by repressing expression of a positive regulator that is required for GBF1 expression. Among the positive regulators of TIA genes tested, only expression of ORCA2 and ORCA3 were found to exhibit a negative correlation with expression of a ZCT gene. Expression of ORCA2 was negatively correlated with ZCT3 expression and expression of ORCA3 was negatively correlated with ZCT2 expression (Table 1). These findings raise the possibility that ZCT2 and ZCT3 could affect GBF1 expression indirectly by repressing expression of ORCA3 and ORCA2, respectively. However, previous results indicating that neither ORCA2 (Li et al., 2013) nor ORCA3 (Peebles et al., 2009) regulate GBF1 transcript levels argue against the possibility that the ZCTs regulate GBF1 indirectly via effects on ORCA2 and/or ORCA3 expression.

GBF1 could act directly to repress expression of specific TIA biosynthetic genes by binding elements in their promoters, thereby inhibiting expression of those genes. Alternatively, GBF1 could inhibit expression of a positive regulator that is required for expression of the TIA biosynthetic genes. Among the positive regulators of TIA gene expression tested, BIS1, CrMYC1 and CrWRKY1 exhibit statistically significant positive correlations with expression levels of all three ZCT gene (Table 1). These findings raise the possibility that GBF1 could repress expression of some TIA biosynthetic genes by turning off expression of BIS1, CrMYC1 and/or CrWRKY1. However, previous research findings indicate that BIS1 does not affect expression of TDC, STR or T19H (Schweizer et al., 2018), indicating that GBF1 does not regulate expression of at least those TIA genes via effects on BIS1 expression levels. In contrast, CrMYC1 has been shown to bind the STR promoter (Chatel et al., 2003), leaving open the possibility that GBF1 could help regulate expression of some TIA biosynthetic genes via effects on CrMYC1 activity levels. CrWRKY1 acts as a positive regulator of TDC, but not of CPR or STR (Suttipanta et al., 2011). Consequently, it is possible that GBF1 could regulate expression of some, but not all, TIA biosynthetic genes via effects on CrWRKY1 expression levels.

Conclusions

Two C. roseus transgenic lines expressing the XPZ RNAi construct exhibit significant decreases in transcript levels of ZCT1, ZCT2 and ZCT3. As the ZCTs are believed to function as repressors of transcription, decreased expression of the ZCTs may be expected to result in increased expression of genes regulated by the ZCTs. However, of eight genes from the TIA and TIA-feeder pathways tested, only expression of T19H increased in the ZCT RNAi lines relative to the control line. This finding raises the possibility that the ZCTs act directly to inhibit expression of T19H, although it is also possible that the ZCTs could act by turning down expression of a positive regulator of T19H expression. In contrast, expression of the other seven TIA biosynthetic genes tested exhibited a positive correlation with the expression levels of all three ZCT genes (Table 1). These findings suggest that the ZCTs affect regulation of these genes indirectly, possibly by turning off expression of a negative transcriptional regulator of these genes. In fact, GBF1 transcript levels are much lower in the control transgenic hairy root lines than in the RNAi lines. This finding suggests a model where all three of the ZCTs act as negative regulators of GBF1 expression and GBF1, in turn, acts as a negative regulator of expression of CPR, LAMT, TDC, STR, 16OMT, D4H and DAT.

Supplemental Information

Supplemental Information 1 Raw data for qRT-PCR experiments.

Control hairy root cultures carrying the XVE-pOpOff2 empty vector and the XPZ28 and XPZ38 hairy root cultures carrying the XPZ RNAi construct designed to reduce expression of all three ZCT genes were grown on media supplemented with either 0 or 20 µM beta-estradiol for the indicated times. Three separate cultures (biological replicates) were grown and harvested for each hairy root line, media and time point. RNA was isolated from each tissue sample and cDNA generated from those RNA preparations. Each cDNA preparation was used as the starting for two PCR reactions, so that there were two technical replicates for each of the three biological replicates for each hairy root line, culture condition and timepoint. The crossing point (Cp) is given for each of these PCR reactions.

Click here for additional data file.

Supplemental Information 2 Schematic of XPZ RNAi construct.

The orientations of the T-DNA right border (RB), fragments of the ZCT1, ZCT2 and ZCT3 genes (ZCT1-3), spacer present in the original pOpOff2(Hyg) RNAi vector (INT), XVE regulatory sequences (XVE), Cauliflower mosaic virus 35S promoter sequences (35S), hygromycin resistance gene (HYG) and T-DNA left border (LB) are indicated.

Click here for additional data file.

Mr. Alex Leopold is thanked for assistance in generating and maintaining the C. roseus transgenic hairy root lines used in these studies. Prof. Jackie Shanks (Iowa State University) is thanked for many interesting and insightful discussions.

Additional Information and Declarations

Competing Interests

Author Contributions

Data Availability

Susan I. Gibson is an Academic Editor for PeerJ.

Chun Yao Li conceived and designed the experiments, performed the experiments, analyzed the data, authored or reviewed drafts of the paper, and approved the final draft.

Susan I. Gibson conceived and designed the experiments, analyzed the data, prepared figures and/or tables, authored or reviewed drafts of the paper, and approved the final draft.

The following information was supplied regarding data availability:

The raw data are available in the Supplemental File.

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
