# Peer review of "Repression of ZCT1, ZCT2 and ZCT3 affects expression of terpenoid indole alkaloid biosynthetic and regulatory genes"

_PeerJ, doi:10.7717/peerj.11624_

## Round 0.1 · original submission · Minor Revisions

Three reviewers brought up various excellent suggestions to improve this manuscript.

Reviewer 1 ·

Basic reporting

no comments.

Experimental design

no comments.

Validity of the findings

no comments.

Additional comments

The manuscript reports RNAi-mediated silencing of three zinc-finger regulators, ZCT1, ZCT2 and ZCT3 in Cathranthus hairy roots and its effect on the expression of selected TIA pathway genes encoding enzymes (TDC, STR, CPR, LAMT, T19H, D4H, DAT, 16OMT) and regulators (ORCA2, ORCA3, BIS1, CrBPF, CrMYC1, CrMYC2, CrWRK1, CrWRKY2, GBF1 and GBF2). Over all the manuscript is well-written, experiments are well-designed with appropriate controls, and the results are well described and interpreted. The manuscript provides some interesting information on the regulation of TIA pathway.

I have a few minor comments.

Introduction:
Pl. clarify. CrBPF1 overexpression alters the expression number of TIA pathway genes but not STR in C. roseus hairy roots. Is it correct?

Materials & Methods:
Pl. described surface-sterilization of seeds in few lines.

line 153: seeds were kept in dark for two weeks?

Pl. include a schematic diagram of the RNAi construct in supplement.

Agrobacterium rhizogenes is widely used to generate hairy roots in a wide range of plant species including Catharanthus? Why did the author choose to use A. tumefaciens GV3101 with separate plasmid with rol genes? Was it more efficient (transformation efficiency etc.) than the conventional method? Pl. clarify.

beta-estradiol was added to 28-day-old culture?

Line 223 RNA extraction and qRT-PCR: It should be Reverse transcription quantitative PCR (RT-qPCR) (according to MIQE guideline: https://doi.org/10.1373/clinchem.2008.112797).

Results:
It would be nice to include PCR or RT-PCR data (integration/expression of HYG and rolB/C gene) confirming the transgenic status of the hairy roots used in this study in Supplement.

Did the authors measure any metabolites in control and transgenic hairy roots? If yes, pl. include the information in supplement.

Reviewer 2 ·

Basic reporting

This manuscript shows how knocking down of ZCT1/2/3 impact the expression of TIA biosynthetic and regulatory genes. The manuscript was well written and the data were presented clearly. I have a few comments on the manuscript.
1. Line 66: “than” should be changed to “then”
2. Line 120: the authors should explain what is PRX1

Experimental design

The experimental design is good

Validity of the findings

1. It seems like ZCT1/2/3 have redundant functions. How similar are they? It would be nice if the authors could provide the comparison of the amino acid sequences of ZCT1/2/3.
2. The authors should explain the difference between GBF1 and GBF2. Why do they have opposite expression patterns in the RNAi lines? If they have opposite expression patterns, the claim that ZCT1/2/3 affect the expression of TIA biosynthetic genes via changing GBF1 expression is not very convincing. The authors should give more discussion on this.

Reviewer 3 ·

Basic reporting

The manuscript is sometimes difficult to follow. I recommend the authors to further polish the writing.

The figures are also difficult to understand as each chart contains too many data points, a lot of which are overlapping. I recommend the authors to make separate charts for the C, 20, and 38 lines for each gene.

Experimental design

I recommend the authors to clarify the necessity of using an inducible promoter for RNAi and of doing a time course.

I recommend the authors to state how they calculated ∆∆Ct in Material and Methods.

Validity of the findings

There is no explanation why an inducible promoter is used for RNAi. However, the inducible promoter of the RNAi vector is very leaky. Expression of ZCT1-3 genes in line 28 and 32 are already much lower than the control line without induction. I recommend the authors to comment on the possibility that sustained repression of ZCT1-3 has different effects than short term repression.

---

## Round 0.2 · accepted · Accept

The authors have addressed the concerns by the reviewers and improved their manuscript.